# Zero- to ultralow-field *J*-spectroscopy with a diamond magnetometer
Muhib Omar [1,2,3] ✉, Jingyan Xu [1,2,3] ✉, Raphael Kircher [1,2,3] ✉, Pouya Sharbati [1,2,3], Shaowen Zhang[1,2,3], Georgios Chatzidrosos[1,2,3], James Eills [4], Román Picazo-Frutos[1,2,3], Dmitry Budker [1,2,3,5] ✉, Danila A. Barskiy [1,2,3] ✉ & Arne Wickenbrock [1,2,3] ✉

Nuclear magnetic resonance (NMR) is a powerful tool for probing molecular structure and dynamics, but conventional high-field systems are bulky and suffer from field inhomogeneities. Zero- to ultra-low-field (ZULF) NMR overcomes these limits by exploiting internal spin interactions in a magnet-free, shielded environment. When combined with nitrogen-vacancy centers in diamond, it enables a compact, portable platform with high spatial resolution and broad bandwidth for noninvasive chemical sensing in microscopic volumes and real-world settings. We report detection of zero- to ultralow-field nuclear magnetic resonance (ZULF NMR) signals at frequencies of a few hertz using a diamond magnetometer. The sensing diamond is a truncated pyramid with $180\,\mu m$ height and a $500^2\,\mu m^2$ base. The minimum stand-off distance is <1 mm, and the sensor sensitivity is 13 pT/$\sqrt{Hz}$ at frequencies $f$ above 5 Hz with $1/f$-like behavior at lower frequencies. NMR signals were generated via signal amplification by reversible exchange (SABRE) parahydrogen-based hyperpolarization resulting in zero-field signals at 1.7 Hz and 3.4 Hz corresponding to the expected hetero-nuclear *J*-coupling pattern of acetonitrile. This work demonstrates a magnet-free platform for detecting chemically specific NMR signals paving the way for portable noninvasive diagnostics in microscopic sample volumes for biomedicine, industrial sensing through metal enclosures.

Nuclear magnetic resonance (NMR) is a powerful spectroscopic technique that provides detailed information on molecular structure and dynamics, with applications spanning chemistry, biochemistry, materials science, and medicine[1]. Traditional high-field NMR relies on strong external magnetic fields to interact with nuclear spins, enabling the detection of signals based on the unique resonances of nuclei in a given chemical environment. However, weak signals, especially in portable benchtop NMR spectrometers with limited field strengths, continue to be a challenge. To address this, various techniques have emerged; for example, parahydrogen-induced nuclear polarization, which significantly enhances NMR signals through hyperpolarization. We specifically hyperpolarize acetonitrile using signal amplification by reversible exchange (SABRE)[2].

Zero-to-ultra-low-field (ZULF) NMR offers a spectroscopic approach by leveraging internal spin interactions rather than relying on strong external fields. The resulting spectral linewidth is narrower than in high-field NMR because of the higher magnetic homogeneity of the shielded environment. This enables high-resolution spectroscopy even of inhomogeneous samples in complex environments[2–4].

A sensor to be used for the magnetic detection of ZULF NMR spectra of miniature samples ideally needs to be small (<1 mm) to enable a short stand-off distance, have high sensitivity (in the pT per $\sqrt{Hz}$ range or better), and a wide bandwidth (on the order of 1 kHz). Currently, sensors for ZULF NMR include superconducting quantum interference devices (SQUIDs)[5], atomic magnetometers[6], and magnetoresistive (MR) sensors[7]. An alternative strategy is shuttling the sample, after evolution under ZULF conditions, into a high magnetic field, allowing for detection by inductive coils[2]. However, the need for cryogenic cooling limits the usability of SQUIDs. MR sensors, while potentially enabling higher spatial resolution, present challenges due to their residual magnetic fields, which can interfere with the ZULF conditions required for spectral acquisition[7]. Optically pumped atomic magnetometers (OPM) can, in principle, achieve high spatial resolution on the order of hundreds of micrometers[8], along with sensing bandwidths

[1]Johannes Gutenberg-Universität Mainz, Mainz, Germany. [2]Helmholtz-Institut Mainz, Mainz, Germany. [3]GSI Helmholtzzentrum für Schwerionenforschung GmbH, Darmstadt, Germany. [4]Institute of Biological Information Processing (IBI-7), Forschungszentrum Jülich, Jülich, Germany. [5]Department of Physics, University of California, Berkeley, CA, USA. ✉e-mail: momar@uni-mainz.de; jingyan.xu@uni-mainz.de; rkircher@uni-mainz.de; budker@uni-mainz.de; barskiy@miami.edu; wickenbr@uni-mainz.de

extending from DC to several hundred kilohertz[9]. However, the sensitivity is rapidly lost for small sensor volumes, and the stand-off distance is limited by the vapor cell itself.

This opens an opportunity for nitrogen-vacancy (NV) centers in diamond, which offer superior spatial resolution down to the nanometer scale[10] and comparatively straightforward implementation of wide-bandwidth sensing, spanning from DC to beyond megahertz frequencies[11]. In addition, NV sensors offer several other advantages, including rapid initialization of the electron spin[11], robust performance, and the ability to operate at a broad temperature range from cryogenic (350 mK)[12] to far above room temperature (550 K)[13]. These sensors have already demonstrated their utility in various areas, for instance, biomagnetic signal detection[14], as well as detection of electric fields[15], rotations[16,17], and temperature[18]. An overview of the various sensors and their typical characteristics used in ZULF NMR is presented in Table 1.

In this work, we explore the potential of combining NV-diamond-based sensors with ZULF NMR. We evaluate their benefits and compare them to commercially available optically pumped magnetometers (OPM, QuSpin). Furthermore, we record ZULF spectra at varying distances and low-field precession spectra of SABRE-hyperpolarized acetonitrile at various fields to highlight possible closer stand-off distances and higher bandwidth of the diamond sensor. The combination of the demonstrated experimental techniques is particularly promising for real-world applications, as it enables portable and low-cost devices.

The paper is structured as follows. We first discuss the general principle of zero-field magnetometry based on NV centers in Sec. II, moving on to the description of the experiment in Sec. III and the presentation of results in Sec. IV. Conclusions are drawn in Sec. V, where we also provide an outlook for future developments.

## NV center zero bias field magnetometry

One of the most common protocols in NV magnetometry is optically detected magnetic resonance (ODMR)[11]. In this method, the diamond electron spins are polarized by optical pumping into the $m_s = 0$ state in the ground-state electron spin-1 manifold. A microwave (MW) magnetic field is applied to drive transitions within the NV ground-state manifold. These transitions move the NV spins into one or both of the depopulated $|m_s| = 1$ sublevels, which experience Zeeman frequency shifts under a magnetic field. Transitions can be optically detected[11] as the photoluminescence (PL) is lower for these states compared to $m_s = 0$. For a single NV center, PL reduces by 30%[11].

To extract magnetic field information from detected frequencies, a sufficiently large bias-magnetic field in the order of at least 100 µT is typically applied. Without this bias, the two $|m_s| = 1$ ground-state sublevels remain (partially) degenerate, and the microwave (MW) resonance feature does not shift but instead broadens to first order under the influence of magnetic fields. This renders the standard ODMR protocol magnetically insensitive in shielded environments. However, removing the comparably large bias field simplifies the experimental setup, reduces reliance on bias

field stability, and allows for the study of signals that would otherwise be perturbed by the bias field, such as ZULF J-coupling spectra.

Several techniques have been developed to enable zero-field sensing with NV centers. One approach uses circularly polarized MW fields[19,20] to selectively excite one of the degenerate sublevels. However, this method is technically challenging because of the difficulty of generating circular polarization along all four NV axes and achieving uniform circularity across the entire diamond volume. Another technique leverages cross-relaxation features near zero magnetic field, which are prominent in NV centers without preferential orientation[21].

However, this method has not yet reached sensitivities comparable to standard finite-field ODMR. Magnetically sensitive noise floors have been demonstrated on the order of around 100 nT per $\sqrt{Hz}$ with 2 nT per $\sqrt{Hz}$ photon-shot noise[22]. The method still requires the application of a DC or modulated bias field. Hyperfine coupling with $^{13}C$ nuclei has also been used for zero-field sensing in single NV centers[23]. However, extending this to NV ensembles requires the use of isotopically enriched $^{13}C$ diamond, which increases electron spin dephasing and degrades sensitivity. Furthermore, ensemble averaging over varying NV-$^{13}C$ distances introduces inhomogeneous effective fields that complicate interpretation.

In this study, we employ a zero-field magnetometry protocol with a modulated bias (6.12 kHz along [100] crystallographic direction with µT amplitude), achieving sensitivities of 13 pT per $\sqrt{Hz}$ comparable to magnetically-biased ODMR[19,24]. This approach provides high-sensitivity sensing without requiring circularly polarized microwaves, as demonstrated and analyzed previously[19]. While the zero-field feature frequency is independent of magnetic field changes around zero bias-magnetic fields, temperature variations still result in frequency shifts[25]. To avoid systematic effects due to these shifts, we stabilize the MW to the zero-field feature. Further details can be found in the "Methods" section.

## Methods
### Hyperpolarization
ZULF NMR signals originate from [$^{15}N$]-acetonitrile molecules. In the absence of a bias field, the main frequency components of the signal are determined by the J-coupling constant of −1.69 Hz (Fig. 1a), resulting in two principal transitions within the zero-bias energy-level structure (see Fig. 1b) of the compound: one at the J-frequency and the other at twice that frequency. These transitions can be magnetically measured and are visible in the amplitude spectrum of the magnetization decay of the sample (Fig. 1c).

Samples were prepared of ([Ir(IMes)(COD)Cl] (IMes = 1,3-bis(2, 4, 6-trimethylphenyl)imidazol-2-ylidene, COD = cyclooctadiene), 5 mM) dissolved in acetonitrile with a stabilizing coligand benzylamine (125 mM). The precursor complex was transformed into the SABRE-active polarization transfer catalyst by dissolving parahydrogen ($pH_2$) at a pressure of 7 bar 15 min prior to detection[3]. The liquid sample was placed in a spherical glass tube, see Fig. 2, and $pH_2$ was bubbled via a glass capillary with a mass flow of 20 scc min$^{-1}$. The $pH_2$ gas handling setup used in this work was described previously[3,4,26]. Acetonitrile solvent was enriched with [$^{15}N$]-acetonitrile to

## Table 1 | Examples of sensor types that detected ZULF NMR

| Sensor | Sensitivity | Bandwidth | Stand-off distance | Dynamic range | Refs. |
|---|---|---|---|---|---|
| NV Diamond | 13 pT per $\sqrt{Hz}$ | ≈3 kHz | <0.2 mm (sensor dimensions) | ≈10 µT | Current work |
| QuSpin OPM | 10 fT per $\sqrt{Hz}$ | 150 Hz | >5 mm | ±5 nT | Current work |
| SQUID | 2.5 fT per $\sqrt{Hz}$[42] | DC-MHz[43] | 30 mm[42] (coil diameter) | ≈100 nT[44] | Matlashov et al.[42] |
| Coil | 20 fT per $\sqrt{Hz}$[42] | 3–10 kHz[42] | 20 mm[42] (coil diameter) | Unspecified | Matlashov et al.[42] |
| OPM | ≈200 fT per $\sqrt{Hz}$[6] | ≈60 Hz[6] | ≈2 mm[6] (cell dimension) | Unspecified | Ledbetter et al.[6] |
| MR | ≈1 pT per $\sqrt{Hz}$[7] | 142.5 Hz[7] | ≈5 mm[7] | ≈100 µT | Picazo-Frutos et al.[7] |

Sensitivity is determined by the magnetically sensitive noise floor of the respective sensor. The stand-off distance denotes a characteristic spatial scale of the sensors. The dynamic range specifies the order of magnitude of the maximum magnetic signal that can be detected by the sensors. The bandwidth corresponds to the frequency at which the magnetic response is reduced by a factor of $\sqrt{2}$ compared to DC for each sensor (or the range specified in the respective paper).

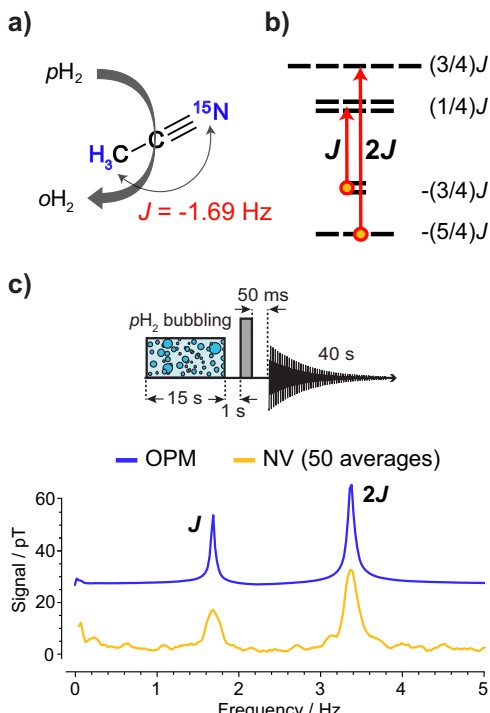

**Fig. 1 | Overview of the experimental scheme and the acetonitrile level structure with an example spectrum. a** Signal amplification by reversible exchange (SABRE) converts the parahydrogen spin order into population imbalance within the $J$-coupled spin states of $^1$H and $^{15}$N in acetonitrile; **b** energy-level diagram of the XA$_3$ spin system of acetonitrile and observable zero-field transitions; **c** experimental zero-field NMR $J$-spectra of hyperpolarized acetonitrile measured with OPM-based (blue) and NV-diamond-based (gold) magnetometers. An offset was applied to the OPM data as a visual aid. The experimental sequence is sketched in the inset. It consisted of 15 s of parahydrogen bubbling, the application of an 860 μs, 6 μT magnetic pulse applied along the $x$-axis (corresponding to the gray box), and 40 s of data acquisition with one of the sensors.

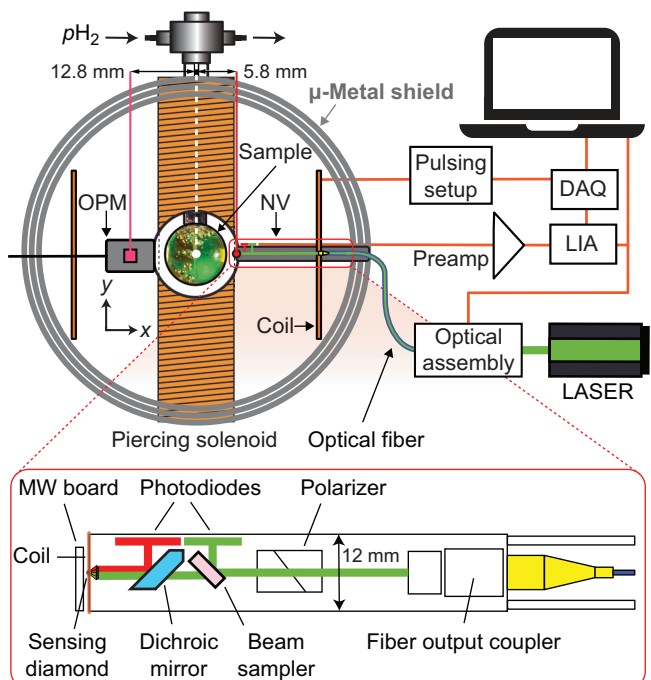

**Fig. 2 | Experimental schematic.** An NMR tube ending in a spherical sample volume, with a parahydrogen bubbling connection. The pink dot in the OPM and the red dot in the NV are illustrating the sensitive volume position in the respective sensor geometry. The data acquisition unit (DAQ), lock-in amplifier (LIA), pulsing setup, and optical assembly are symbolically sketched. The magnetic pulse to initiate a free magnetization decay was applied using the (Helmholtz) coil pair.

25% (natural isotopic abundance of $^{15}$N nuclei in acetonitrile is around 0.4% and was increased with pure [$^{15}$N]-acetonitrile). Hyperpolarization of [$^{15}$N]-acetonitrile can be generated effectively in ZULF conditions via various protocols, in our case, a so-called intermittent bubbling sequence[4]. The sequence is sketched in Fig. 1c.

A pulse of 860 μs with 6 μT amplitude is applied to initiate the free magnetization decay. The diamond sensor data acquisition lasted 40 s, initiated by a software trigger synchronized with the ZULF sequence. All chemicals except for the catalyst were acquired through Sigma-Aldrich. The catalyst was prepared based on previously developed methods[27–29].

### Zero-field sensing and microwave locking

In the following, we discuss the microwave locking technique and the zero-field sensing methodology. To understand the mechanism, it is sufficient to consider the ODMR signal for a single axis, neglecting hyperfine structure under MW driving at frequency $\nu$. With a subtracted background PL, the signal can be modeled as proportional to the sum of two Lorentzian functions. One for each ground-state transition $m_S = 0 \rightarrow \pm 1$:

$$P(\nu) \propto \sum_{m_S=\{-1,1\}} \frac{(\Gamma/2)^2}{[\nu - \nu_0(T) - m_s \nu_L(B)]^2 + (\Gamma/2)^2}, \quad (1)$$

where $\Gamma$ is the full-width-at-half-maximum (FWHM), $\nu_0(T) = D_{300K} + \kappa T$ is the linearly approximated temperature dependent zero-field ground-state transition frequency with temperature independent offset frequency,

$D_{300K}$ = 2870 MHz and proportionality constant $\kappa$. In general, $\kappa$ is temperature dependent[30], but at room temperature, $\kappa \approx -78$ kHz per K[31]. The Larmor frequency $\nu_L(B) = \gamma B$ is a linear shift of the resonance frequency as a function of the applied on-axis magnetic field $B$, which changes sign depending on the $m_S$ level. The NV center's gyromagnetic ratio is $\gamma = 28$ GHz per T.

At $B = 0$ T, the two transitions become degenerate. Phase-sensitive detection of $P$ with a lock-in amplifier (LIA) for a small modulation of the MW frequency results in the derivative signal of Eq. (1) with respect to the microwave frequency $\partial P/\partial \nu$. Without a background magnetic field, this signal is dispersive around $\nu_0(T)$ and can be used to lock the microwave frequency.

The derivative can be evaluated:

$$\left. \frac{\partial P}{\partial \nu} \right|_{\nu_0, B=0} (\nu) = c_1 (\nu - D_{300K} + \kappa T) + \mathcal{O}(\nu^3), \quad (2)$$

where $c_1$ is a proportionality constant depending on the lock-in settings, modulation parameters, and resonance properties. The Zeeman shift cancels itself, and linear sensitivity to temperature is retained. Similarly, for a small modulation of the magnetic field $B$ around zero field, the derivative with respect to the offset magnetic field $\partial P/\partial B$ can be computed using phase-sensitive detection. To first order, this is linearly dependent on the background magnetic field $B$:

$$\left. \frac{\partial P}{\partial B} \right|_{\nu_0, B=0} (B) = c_2 B + \mathcal{O}(\nu^3), \quad (3)$$

where $c_2$ is a proportionality constant depending on modulation parameter, lock-in, and resonance properties. By simultaneously modulating both MW frequency and magnetic field $B$ and demodulating at separate reference frequencies, two independent dispersive signals can be obtained:

one sensitive primarily to MW-frequency shifts (e.g., due to temperature drifts) and one linear in magnetic field. Care has to be taken that the frequencies are sufficiently different, such that there are no cross-modulation features in the frequency range of the magnetometer.

This forms the basis for high-sensitivity zero-field magnetometry using NV centers. In practice, we generate an error signal for the microwave locking by sinusoidally modulating the MW frequency (modulation frequency 140 kHz, amplitude 100 kHz) and demodulating it with a LIA. The magnetically sensitive signal similarly results from sinusoidally modulating the magnetic field (modulation frequency 6.12 kHz, amplitude ≈3 μT) and demodulating it with a second LIA. Stabilizing the MW frequency using a proportional-integral-derivative control loop thus makes the system largely insensitive to temperature fluctuations, enabling long-term measurements, while the output of the second LIA is a direct measure of the magnetic field at the position of the diamond.

## Diamond sensor

The principle of the sensor is based on the ODMR technique[11]. The sensor head is designed to optimize light collection[32], see Fig. 2. It includes an output coupler for the optical fiber delivering the pump laser light, and producing a beam of 300 μm in diameter. This beam passes through a polarizing beam splitter to clean its polarization. A beam sampler captures part of the light before the beam propagates through the backside of a dichroic mirror and into a lens that focuses the light into the sensing diamond (containing 3.7 ppm of NV and <10 ppm of nitrogen). The sensing diamond is shaped like a truncated pyramid for side collection and mounted on a diamond anvil to enhance the collection efficiency by a factor of four[32]. In addition, a silver reflective coating was applied to the diamond components to avoid leakage of the green laser light and to increase light collection efficiency further. The red PL emitted by the diamond is collected with the same lens that focuses the laser beam. This light is reflected by the front side of the dichroic mirror and directed to a photodiode (PD). The sensor uses non-magnetic Hamamatsu photodiodes (S13228-01) soldered onto a printed circuit board (PCB). The current from the green pick-up beam PD and the red light PD are subtracted directly on the PCB for differential detection to reduce the sensitivity to laser intensity variation. A 3D-printed clipping mask is used to partially cover the green pick-up beam PD, allowing fine adjustment of the differential detection scheme[24]. With 150 mW of green laser light power focused on the diamond, 5 mW of red photo-luminescent light is collected on the photodiode. The mobile sensor setup includes an instrument rack on wheels. This rack includes a multi-frequency LIA (Zurich Instruments AG, HF2LI), a breadboard with a Coherent, Verdi G5 laser (532 nm) coupled into a high-power single-mode optical fiber that delivers light to the sensor head, a preamplifier to enhance the signal from the sensor head, and a microwave signal generator. The output of the microwave generator Rohde Schwarz, SMA100b, is mixed with that of an RF source (Keysight Technologies, 33512B) and then amplified by a microwave amplifier (Mini-circuits, ZHL-16W-43-S+). This frequency-mixing technique alters the hyperfine-resolved spectrum by simultaneously driving two resonance transitions with two microwave frequencies and therefore increases the optical contrast of the electron spin resonance. This, in turn, directly enhances the magnetic measurement sensitivity[11]. The microwaves are delivered to the sensor via a flexible PCB. The differential signal of the sensor head is sent to the preamplifier and subsequently to the LIA. Around the diamond, a coil is mounted, and its current is modulated with one of the LIA outputs to enable the magnetic sensing protocol. The custom-designed PCBs for the photodiode and microwave delivery are discussed in the SI.

The diamond sensor signal is acquired via an LIA and recorded on a personal computer (PC). A trigger synchronized to the magnetic pulse, which initiates the free evolution of the sample's nuclear magnetization, is used to start the data acquisition. The resulting voltage time series is averaged 50 times and converted to a magnetic field time series using a calibration factor. The factor was determined by application of a calibrated test signal. A fast Fourier transform (FFT) is then applied to extract the single-sided amplitude spectrum. The FFT scaling is chosen so that a sinusoidal signal with an amplitude of 100 pT produces a corresponding spectral peak of 100 pT. The acquisition rate of the LIA is fixed at 14 k samples per s. The LIA low-pass filter bandwidth is set to 100 Hz, 350 Hz, and 600 Hz, depending on the expected signal frequency. These spectra constitute the J-coupling spectroscopy using the diamond sensor, shown for instance in Fig. 1c. The transitions of the coupled spin system appear as peaks in the amplitude spectrum.

In order to quantify the sensitivity of the diamond sensor, we acquire 9 s data traces. The corresponding amplitude spectrum needs to be rescaled to an amplitude spectral density (ASD) to characterize the noise floor. This requires division by the square root of the effective noise bandwidth[33], in this case, 0.17 Hz, considering a Hann window and 9 s duration. We record magnetically sensitive, insensitive, and electronic noise spectra. The recorded magnetically sensitive spectral densities are acquired while magnetic field modulation and light are on, the insensitive spectral densities while the magnetic field modulation is off and light is on, and the electronic noise spectral densities while both light and magnetic modulation are turned off. The resulting characterization noise spectral densities are shown in Fig. 4i. The magnetically sensitive noise floor features a 1/f-noise profile and matches the magnetically insensitive noise floor above about 5 Hz. The average of the magnetically insensitive noise spectral density below 50 Hz is 13 pT per $\sqrt{Hz}$.

## The vapor cell magnetometer

The optically pumped magnetometer is a commercially available hot vapor cell-based sensor (QUSPIN) employing rubidium atoms to measure the magnetic-field-dependent absorption of laser light[34,35]. The bandwidth of the sensor is specified as 150 Hz, the dynamic range as ±5 nT, and the sensitivity on the order of 10 fT per $\sqrt{Hz}$[34]. The OPM has a sharp hardware digital low-pass filter at 500 Hz[34], which inhibits sensing fields beyond that frequency. The OPM signal was recorded after initialization by its commercial software and read out with a custom-made Python code. The OPM data is acquired without averaging.

## Experimental arrangement

The setup is sketched in Fig. 2. The experiment is conducted inside an MS-2 magnetic shield from Twinleaf. The OPM sensor is mounted within a 3D-printed holder; it is sensitive along the same axis as the diamond sensor (x-axis). OPM and diamond sensor are positioned on the x-axis around the spherical NMR sample (12.5 mm diameter). Both sensors are inside a Helmholtz-coil pair oriented along the sensing direction, which provides the excitation pulse to initiate the free decay of the NMR sample magnetization. The NMR sample is enclosed within a solenoid wound around a ceramic tube (a "piercing" solenoid)[36] that allows applying magnetic fields along the y-axis without disturbing the OPM sensor. This arrangement allows observation of nuclear spin precession around the y-axis within the x-z plane during the field precession experiments.

## Results

We measure NMR spectra of acetonitrile with the diamond sensor at various distances and piercing solenoid fields, and compare the results with both simulations and the OPM sensor data. We aim to showcase potential advantages of the NV sensor due to the smaller sensing volume and the sensor design: it features a wider bandwidth and allows smaller sample-sensor distances compared to the OPMs. The OPM and diamond measurements were conducted alternatively, as the magnetic field modulation of the diamond measurement protocol affects the OPM performance.

The diamond ZULF NMR spectra obtained with the NV-based sensor are shown in Fig. 3. We compare them with OPM measurements and with simulation results at zero field and an applied field along the y-axis of 10 nT and 40 nT. The numerical simulations take into account the hyperpolarization arising from the bubbling procedure, the application of a pulse along the x-axis, and the bias field application along the y-axis. At zero field, the spectra originate primarily from scalar (J) couplings among the $^1$H-$^{15}$N

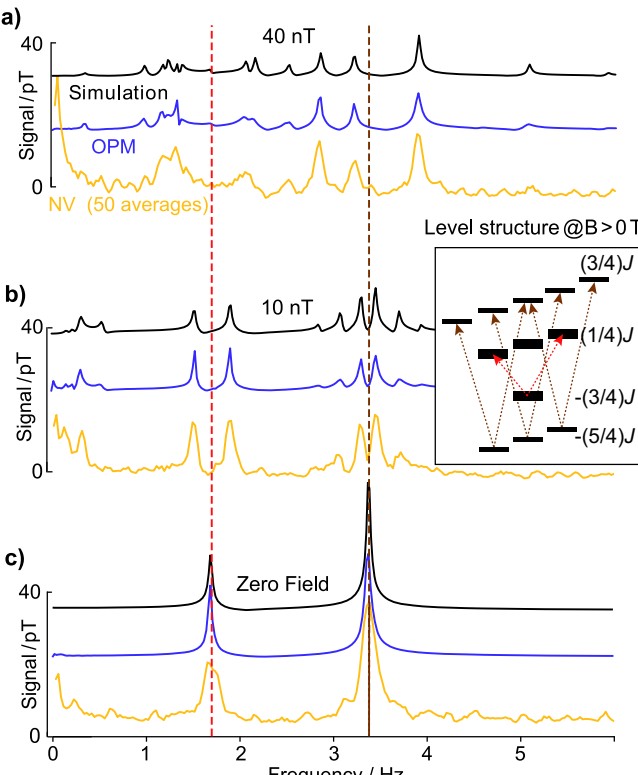

**Fig. 3 | Comparison of the measured and simulated J-coupling spectra of [15N]-acetonitrile at different ZULF fields. a** 40 nT, **b** 10 nT, and **c** 0 nT. OPM (blue) in the middle, diamond sensor (gold) at the bottom, and simulation (black) at the top. Diamond data is taken at 13.3 mm distance from the NMR sample center, and Fourier transformed after 50 averages in the time domain. Dashed lines indicate the 1.7 Hz ($J$) and 3.4 Hz ($2$-$J$) frequencies. Note that for each field, the simulation, OPM data, and NV data are shifted as a visual aid. The allowed transitions due to the adjusted selection rules for (**b**) under a 10 nT transverse magnetic field (to the quantization axis) are sketched in the level structure as an inset. For (**c**), the energy-level structure becomes more complicated. The minor disagreement between simulation and experimental data can be explained with experimental imperfections like magnetic field gradients of the applied fields and the noise properties of the sensors, in particular, the 1/f noise of the diamond sensor.

nuclei in acetonitrile. At ultra-low fields, where the Zeeman energies are comparable to the J-coupling strength (on the order of tens of nanotesla), the spectra exhibit more complicated patterns in good agreement with numerical simulations. This demonstrates that the sensor and the implemented sensing method is compatible with the acquisition of ZULF NMR.

The measurement sequence to showcase the bandwidth advantage of the diamond sensor compared to the OPM sensor is presented in Fig. 4a–f. We measure NMR spectra with an OPM and diamond sensor for various magnetic background fields up to 14 µT. The diamond sensor records NMR features up to 580 Hz. The OPM does not measure signals beyond 500 Hz.

We observe broadening of the spectra as a function of the applied field. We relate this to inhomogeneities of the piercing solenoid field. The signals recorded with the diamond sensor are slightly larger than the OPM's, which is due to the slightly closer distance of the diamond sensor to the NMR sample. Additionally, we record $^{15}$N-related precession resonances [Fig. 4c, d] with the OPM. The signals (maximum amplitude around 1 pT) are much smaller than the J-coupling spectra (maximum amplitude 35 pT). In the OPM spectra at 14 µT, the $^{15}$N resonances are not resolvable since they partially overlap with artifacts due to the 50 Hz power line. In Fig. 4g, the mean frequency shifts of the NMR $^{1}$H/$^{15}$N features are displayed and fitted linearly. The slopes of the fits are (41.1 ± 0.1) Hz per µT for $^{1}$H precession and (−4.1 Hz per µT) for $^{15}$N-precession. This is close to the expected

gyromagnetic ratio for free $^{1}$H (42.57 Hz per µT) and $^{15}$N (−4.3 Hz per µT). In the diamond data, no $^{15}$N-precession features are resolvable. We relate this due to the sensitivity difference of the sensors. This can be further illustrated in two, ideally complementary, ways. Both the sensitivity characterized by the ASD, as well as the histogram of the amplitude distribution of the diamond sensor data, will be used in the following to discuss this limitation quantitatively. The ASD of the diamond sensor noise is displayed in Fig. 4i. We measure 9 s time traces of either magnetically sensitive (i.e., with magnetic field modulation and pump light applied), insensitive (i.e., with only pump light applied) and dark (i.e., with neither pump light nor magnetic field modulation applied) diamond data and apply a Fourier transform to extract the ASD as a characterization of the magnetic sensitivity. The magnetic sensitivity of the diamond sensor, given by the noise baseline of the magnetically sensitive ASD of the sensor, is measured to be 13 pT per $\sqrt{Hz}$ (ASD average within the range: 20–30 Hz). The expected amplitude spectrum root mean square (RMS) value for the single-sided ASD of 13 pT per $\sqrt{Hz}$, considering the applied Hann window before the Fourier transform, 9 s measurement time, and 50 averages, is 0.8 pT.

In Fig. 4h, we show the histogram of the diamond sensor amplitude spectrum between 26 and 35 Hz. The histogram data is Rayleigh distributed and fitted accordingly. A Rayleigh-distributed histogram is expected for the absolute value of the single-sided amplitude spectrum of normally distributed random time series data. The fitting parameter of the Rayleigh distribution $\sigma$ is 1.4 pT. The corresponding RMS is $\sqrt{2}\sigma \approx 2$ pT, larger than the $^{15}$N-related precession feature amplitude.

The value derived from the ASD is roughly a factor of 2 smaller than that derived from the histogram of the amplitude spectrum. We attribute this discrepancy to a residual effect of the 1/f tail already visible in the magnetically sensitive ASD in Fig. 4i. The 40 s long time traces used in the amplitude spectra are more sensitive to 1/f related effects compared to the 9 s trace used for the sensor characterization.

The scaling of the spectra measured with the diamond sensor as a function of distance is shown in Fig. 5. We expect the signal to follow an inverse cubic distance law, assuming the magnetic field of the NMR sample can be approximated with a single dipole. This scaling behavior is observed for distances from 9.3 mm sensor volume center to sample volume center. For closer distances of the diamond sensor, the spectra change. The signal amplitude appears smaller than expected from dipolar scaling and qualitatively different, akin to spectra of samples subjected to a magnetic field or a magnetic field gradient.

## Discussion and conclusion
In this work, we demonstrate a magnetometer based on an ensemble of NV centers in diamond compatible with operation without a background field. Key elements of this sensor are minimal sensor-sample distances of below 0.2 mm and a sensitivity of 13 pT per $\sqrt{Hz}$ (photon-shot noise limited 2 pT per $\sqrt{Hz}$). This sensitivity is sufficient to detect NMR spectra under zero and ultralow field conditions of hyperpolarized samples. We discuss potential avenues for improving the sensor performance by analyzing the noise characteristics of the diamond magnetometer. Following a discussion of the distance-scaling measurement results, we highlight the expected advantages in miniaturization enabled by the diamond sensor and conclude with a summarizing outlook.

The high sensitivity of the diamond sensor could be further improved. Currently, the magnetic noise floor of the sensor [in Fig. 4i] is dominated by 1/f noise at frequencies below 5 Hz (for 9 s data traces). A relevant frequency range, since some of the observed NMR signals occur there. The magnetically insensitive noise trace does not show this low-frequency noise, indicating that neither laser nor electronic noise is responsible. Environmental magnetic noise can also be excluded, as the OPM sensor does not exhibit comparable features (see Fig. 4 in the SI). Temperature effects that could mimic such a noise increase are mitigated using a microwave frequency lock. Therefore, a plausible explanation is that the observed 1/f noise originates magnetically from the field modulation source of the diamond sensing protocol. This is inactive during OPM operation. The DC component of the

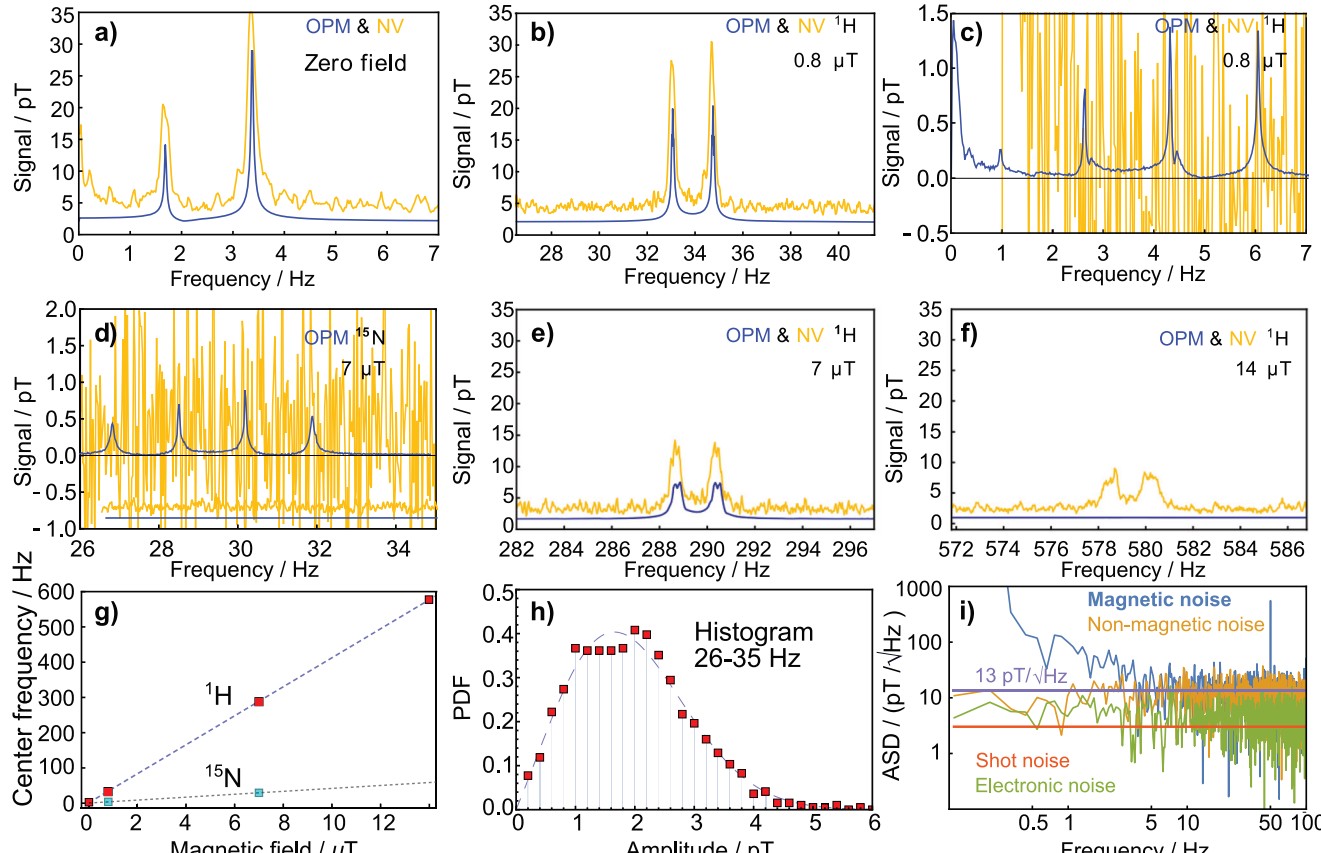

**Fig. 4 | Scaling of the NMR spectra as a function of background magnetic field and sensor characterization.** Spectra (**a–f**) are taken while applying fields on the NMR sample using the piercing solenoid. The [1]H-dominated spectra measured using the OPM are shown in blue and using the diamond sensor in gold. NV data is 50 times averaged in the time domain before Fourier transforming it into an amplitude spectrum. The NV data in this figure are all recorded at a distance of 12.8 mm from the NMR sample center. The OPM data show no response above 500 Hz. [15]N-related precession resonances in acetonitrile measured with the OPM sensor at 0.8 µT and 7 µT are not detectable with the NV sensor within 50 averages. In (**c, d**), the diamond data are vertically offset to enable a clearer comparison with the OPM data. The mean signal level in (**d**), for instance, without this offset is approximately 1.5 pT [see **h**]. **g** Mean center frequency of all detected [1]H- and [15]N-dominated resonances at various bias fields using the OPM data. The [15]N-dominated resonances are not resolvable at 14 µT. **h** Histogram of the diamond sensor amplitude spectrum in the range of the [15]N-dominated resonances at 7 µT [from **d**] displayed as a probability density function (PDF) and fitted with a Rayleigh distribution. **i** Noise spectral density of the diamond sensor to characterize its sensitivity inside the magnetic shield. The non-magnetic noise floor was recorded without magnetic field modulation, and the electronic noise floor without laser light.

source is necessary to calibrate the sensitivity of the sensor by applying constant fields of known amplitude. In a future iteration of this sensor, the calibration protocol will be modified and the modulation coil current high-pass filtered, effectively removing this 1/f-noise contribution. At higher frequencies, the magnetically insensitive noise floor dominates, indicating laser-noise-limited sensor performance. An improved laser noise cancellation circuit could further reduce this, ideally approaching the shot-noise-limited sensitivity of approximately 2 pT per $\sqrt{Hz}$.

We demonstrate that the diamond sensor design allows minimal sample-sensor distances of below 0.2 mm corresponding to the measurement trace labeled 6.8 mm sample center to sensor center distance in Fig. 5d. The scaling of the integrated signal strength as a function of distance from the spherically shaped NMR sample is shown in Fig. 5a. The signal strength is derived by numerically integrating the amplitude spectrum around the 2$J$ resonance from 2.5 to 6 Hz. From this number, we subtract the numerical integral of the amplitude spectrum over a similar frequency range containing no signal (from 6.2 to 9.7 Hz). The results are normalized to the values at a sensor-sample distance of 14.3 mm for both sensor modalities. The signal scaling is compared to the expected dipolar scaling of a uniformly magnetized sphere. The relative signal strength increases with decreasing distance for both sensors. The maximum diamond signal strength is three times larger at closer distances compared to the signal strength at 14.3 mm.

Both the OPM and diamond data follow the expected curve at larger distances, but deviate from it at shorter distances. The deviation of the diamond sensor is much more visible since it is moved much closer to the NMR sample. Broadening of the spectra can be a reason for this apparent suppression of signal strength increase.

The diamond-detected spectra for close sample-sensor distances appear broadened. Surprisingly, with both sensors operating under identical conditions around the NMR sample, the OPM data appear far less broadened. Both the OPM and the diamond sensor are turned off when the other sensor is in operation to avoid interference from each sensor's modulating magnetic field. Broadening of NMR spectra can be related to varying transverse magnetic fields during the measurement time and/or magnetic field gradients over the sample.

Possible sources of magnetic fields and gradients are investigated. We find that the diamond sensor components are slightly magnetic. Spectra measured with an OPM for different diamond sensor components placed close to the NMR sample can be found in SI, Supplementary Fig. 4. The broadening could also be caused by slow magnetic field drifts of the modulation source that could affect the NMR spectra.

The interpretation of the broadening is complicated by the shimming procedure used during the sensor distance experiment. Shimming is conducted for each diamond sensor position using the OPM sensor recorded

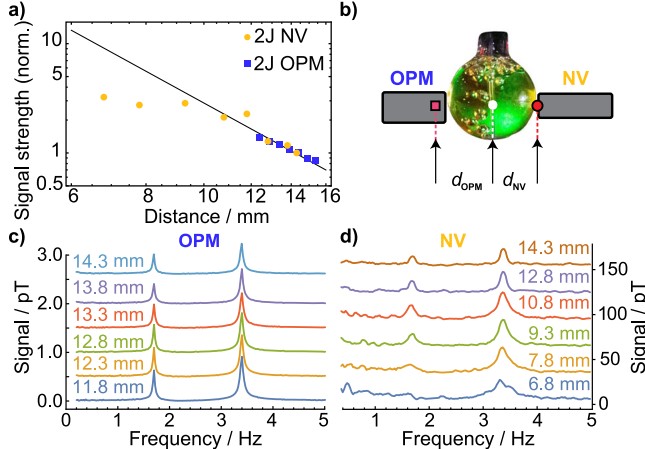

**Fig. 5 | Distance scaling of the NMR spectra.** Distance is given from the sensor volume center to the sample volume center. The sample is a glass sphere with a 6.25 mm radius surrounded by a piercing solenoid with a 0.5 mm wall thickness. **a** Normalized signal strength of the 2J-peak measured at different distances with respect to the NMR sample for the diamond (gold) and OPM (blue) sensor. The signal strength is the baseline-subtracted numerical integral of the spectra between 2.5 and 6 Hz, capturing the 2$J$ peak for the respective sensor. For each sensor, the signal strength is normalized to the value at a distance of 14.3 mm. The closest distance possible for the OPM sensor is 11.8 mm. The closest distance for the diamond sensor is 6.8 mm, dominated by the diameter of the piercing solenoid. The normalized signal strength is compared to the expected scaling for a dipole source (black). **b** Sketch of the experiment. **c** Spectra recorded using the OPM for various distances. **d** Spectra recorded using the NV sensor for various distances. NV data are 50 times averaged in the time domain before Fourier transforming. The spectra in (**c**, **d**) are offset as a visual aid. Note that the OPM data are recorded with natural abundance acetonitrile, leading to the much smaller signal size compared to the diamond data.

spectra as a reference. This is necessary to compensate for the effect of the slight magnetic field components of the diamond sensor. In a future iteration of the experiment, reducing the magnetic influence of the diamond sensor using fewer magnetic components and a high-pass filtered current source might resolve that discrepancy.

In summary, NV-diamond sensors offer unique advantages in the context of ZULF NMR. One such advantage lies in their compact form factor, which allows positioning of the sensor in close proximity to the sample—within hundreds of microns—enabling high spatial resolution and the investigation of smaller sample volumes (on the order of the diamond sensor dimensions). The demonstration of ZULF NMR in this manuscript confirms this possibility, once the sensor's residual magnetic field effects are under control. These can be expected to be less of a problem for smaller samples. Sample volumes on the order of the diamond sensor dimensions are possible to detect, since the surface magnetic field of a uniformly magnetized spherical sample is independent of its radius $r$. The field outside the sample is equivalent to the field of a magnetic point dipole at the center of the sample with a magnetic moment proportional to the number of spins. For a constant density, the number of spins scales with the volume ($r^3$) while the signal of a dipole measured at position $r$ reduces like $r^{-3}$. Therefore, in the regime where the sample magnetization can be treated as a classical dipole, miniaturization of the sensor permits probing smaller samples without a fundamental loss in sensitivity.

The high bandwidth of the diamond sensor is useful to study $J$-spectra of molecules with large $J$-coupling. Platinum $J$-couplings, for example, have been calculated to lie in the range of tens of kHz[37] and $PF_6^-$ exhibits $J$-couplings of several multiples of 700 Hz[38,39].

Similarly, the study of the NMR spectra in the intermediate or uncoupled regimes is more easily accessible with higher bandwidth, providing the most complete information about the spin system (including both $J$-couplings and chemical shifts)[40]. We demonstrate the advantage of

the higher bandwidth of the diamond sensor in Fig. 4f. At fields where the precession frequency exceeds 500 Hz, no proper response from the OPM can be obtained as it goes beyond the 6th order hardware low-pass filter[34].

In the following, we discuss the generalization of our results to other ZULF NMR samples. Firstly, at zero field, a source of polarization is always required to produce detectable signals[2]. Among the various techniques available[41], SABRE is selected in this study due to its ability to produce high molar polarization under ambient conditions. However, the methodology presented here is not limited to SABRE. One can also employ several complementary hyperpolarization methods, including high-field pre-polarization followed by sample shuttling, dynamic nuclear polarization (DNP), photo-induced DNP, and other parahydrogen-based approaches. While SABRE enables the first demonstration to our knowledge of diamond-based ZULF NMR, these alternative strategies broaden the scope of detectable molecular targets, including those not amenable to parahydrogen-based hyperpolarization. Thus, we view the use of NV sensors in conjunction with diverse hyperpolarization techniques as a viable platform for scalable and versatile low-field NMR detection for a variety of polarizable compounds.

The demonstrated combination of NV-diamond magnetometry and hyperpolarization-enhanced zero-field NMR paves the way to highly integrated quantum diagnostics in real-world environments without the requirement for large shields and without the high cost of superconducting magnets.

## Data availability
The data supporting the findings of this study are included in the paper and available from the corresponding https://doi.org/10.6084/m9.figshare.30752147.

## Code availability
The programming codes related to data analysis and control of the OPM sensor are available from the corresponding authors upon reasonable request.

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

## Acknowledgements

This work was supported by the EU, project HEU-RIA-MUQUABIS-101070546, the Initiative and Networking Fund of the Helmholtz Association (Project No. VH-NG-20-20) by the Helmholtz Association project Quantum Sensing for Fundamental Physics (QS4Physics) from the Innovation pool of the research field Helmholtz Matter, by the DFG, project FKZ: SFB 1552/1 465145163, by the Alexander von Humboldt Foundation in the framework of the Sofja Kovalevskaja award and by the German Federal Ministry of Research, Technology and Space (BMFTR) within the Quantumtechnologien program via the DIAQNOS project (project no. 13N16455). The authors thank Hamamatsu for providing the photodiodes as samples.

## Author contributions

Conceptualization: D.B., D.A.B., and A.W. Methodology: D.B., D.A.B., R.K., R.P.F., G.C., J.E., and A.W. Acquisition of experimental data: M.O., J.X., R.K., and P.S. Visualization: M.O., J.X., R.K., S.Z., R.P.F., G.C., D.A.B., and A.W. All authors contributed to the writing, read, and edited the manuscript.

## Funding

## Competing interests

The authors declare no competing interests.
