## [Transparent Peer Review file · Communications Chemistry]

Zero- to ultralow-field J-spectroscopy with a diamond magnetometer

Corresponding Author: Mr Muhib Omar

Version 0:

Reviewer comments:

Reviewer #1

(Remarks to the Author)

The manuscript entitled 'Zero- to low-field J-spectroscopy with a diamond magnetometer' demonstrates the advantages of diamond magnetometers for conducting ZULF (Zero- to ultralow-field) NMR spectroscopy. Most of results are about the comparisons between a commercial optically pumped magnetometer (OPM) and a home-built diamond magnetometer. Obviously, OPM provides a way higher SNR due to its high-sensitivity. However, the diamond magnetometer has several advantages over OPM in terms of bandwidth and detection volume. The results sound reasonable. The design of the home-built diamond magnetometer looks highly practical.

This work is indeed the first demonstration of NV diamond detected ZULF-NMR. However, I doubt that it actually demonstrates significant advances. The feature that NV diamond can do sensing at over 500 Hz seems a minor factor, compared to the OPMs high-sensitivity. To achieve the same SNR as that with OPM, NV diamond may need hundreds of scans, which is certainly a significant disadvantage. Moreover, the demonstration of the 2nd advantage that NV diamond can detect small volume samples is not clear in this manuscript. The authors could have used unconventional samples with sub-millimeter scales such as micro-fluids, where OPM detected ZULF NMR is nearly impractical. In addition, I noticed that some of important technical concepts are not clearly described in the manuscript. Thus, not considering the scientific impact of the manuscript, I don't recommend it for publication in Communications in Chemistry.

Below are the comments enquiring more explanations.

1. The authors stated they used a modulated bias protocol for the operation of their diamond sensor for zero-field magnetometry. But, it's is rather hard to understand the working-principle and real implementation of this protocol. The authors should include a more detail description on it.
2. There are several wrong citation of figures. For example, in page 4, "the diamond sensor compared to the OPM sensor is presented in Fig. 3", Fig.3 -> Fig. 4. In page 5, "the advantage of the higher bandwidth of the diamond sensor is in Fig. 3.", Fig.3 -> Fig.4.
3. Please clarify the materials that the authors used for NV diamond sensor and the potential source of the residual magnetic field gradient.
4. Please explain why with NV diamond sensor the ^{15}N NMR spectra were invisible.

Reviewer #2

(Remarks to the Author)

Muhib Omar et al. report on low-field detection of nuclear magnetic resonance with ensembles of nitrogen vacancy centers. They focus their attention on the measurement of the ZULF-NMR spectrum at (or close to) zero magnetic field, and together with the comparison with the OPM measurements, they are able to validate the multi-peak spectrum resulting from a combination of J-coupling and small bias magnetic fields. Not only that the manuscript has quite a few novelties, its high quality and scientific rigor merit publication in COMMSCHEM. I really enjoyed reading it.

However, I would like to point out to the authors that there are many sloppy errors in the manuscript which indicates that the manuscript was not carefully proofread by the authors before submission. This will increase the time the reviewers have to

spend to understand all the details.

There are several points the authors should address before proceeding to publication. Please see below:

Major comments for improving the presentation of the manuscript:

1. Since the diamond sensor, flexible PCB and other components of the experimental setup used in this work were not described in previous papers, a detailed description containing the part numbers of the components used should be included in the section IIIB. An actual photograph of the diamond sensor setup should be included in the Supporting Information document.
2. A Table summarizing the comparison of diamond magnetometer with other ZULF-NMR sensors such as OPMs, SQUIDs and magnetoresistive sensors etc. should be included. The parameters to be considered are sensitivity, dynamic range, bandwidth, stand-off distance, blocked optical access etc. The corresponding References should also be cited in the Table.
3. Page 4, 3rd Paragraph: The authors mention "Although this demodulated signal is magnetically insensitive to first order (as discussed earlier)". It is not clear to me in which section was it discussed earlier. In fact, the entire paragraph is not clear to me. The authors should support their statements by analytical expressions and they should be included in the Methods section.
4. Please clarify why the Fourier transform (FT) is applied after 50 averages in the time domain. Why not average the 50 FTs?
5. Please clarify in the text why "the 15N resonances in acetonitrile were detectable only with the OPM." using sensitivity calculation arguments, if possible.
6. If I am not wrong, the data in most of the subfigures of the Figure 4 is not consistent with the description given in the caption. Please double-check.
7. Please specify the sources of the "1/f"-like magnetic noise in the Figure 4(i) and how the noise can be mitigated. Please cite the references, if possible.
8. Please specify in the text the details of the equivalent noise bandwidth and the hardware low-pass filtering used for the diamond sensor.
9. "The sensitivity was sufficient to resolve the characteristic lines at zero field in the diamond data related to 1H-15N nuclei in isotopically labeled acetonitrile within 50 averages." I got an impression that 15N resonances are not detectable?

Minor comments and corrections:

1. Section IIIA, 1st Paragraph, 2nd Sentence: "prior to" should be more appropriate. The same sentence, which ends with both full stop and a comma, should be corrected.
2. Green laser excitation power used for ZULF NMR measurements should be mentioned.
3. Page 4, Paragraph 4: "We average the noise floor between 5 and 30 Hz." This needs to be corrected.
4. Page 4, Paragraph 4: "turning off" instead of "turning of".
5. The frequency of the modulated bias field and the MW field should be mentioned.
6. The title for the inset of the Figure 3 is not completely visible.
7. Figure 3: Dashed lines instead of dotted lines.
8. The description of the Figures in the main text and the corresponding Figure numbers are not consistent with each other at several places in the manuscript.
9. Page 4, Section IIID: "For observing the nuclear spin precession along the y-axis". Please specify the plane in which the transverse nuclear magnetization precesses.
10. Please double-check for any errors in the caption of figure 5.
11. Section V, 1st Paragraph: Please check for the presence of any errors.

I request the authors to address the above points in the revised version of the manuscript. The manuscript can be sent to me for review again.

Reviewer #3

(Remarks to the Author)

The paper is a demonstration of zero/ultra-low field (ZULF) NMR magnetometry with nitrogen-vacancy (NV) centers in diamond, as a means of measuring J-coupling strengths. The authors built an apparatus to perform ZULF NMR spectroscopy with the NV center as a probe and compared the NV magnetometry results to results obtained through commercial techniques. The authors examine [15N]-acetonitrile using PHIP-SABRE (a para-hydrogen based nuclear spin hyperpolarization technique) and measure the hyperpolarized J-coupling signal from the sample using ODMR. The authors compare the efficacy of the diamond sensor to a standard vapor-cell-based optically pumped magnetometer (OPM) method, and demonstrate that though the diamond sensor does not have a comparable noise floor to the OPM, it can currently be useful at frequencies above the low-pass cutoff of the OPM.

This is a very nice demonstration of NV-based NMR in the ZULF regime. This novel result, and the comparison with OPM methods could be of interest to the community and the wider field. It is a little harder to gauge the significance of the results presented. Generally, NV magnetometry is known to have better spatial resolution but worse magnetic field sensitivity compared to an OPM. The comparison between the NV and OPM platform ideally needs to be performed in the context of the target application. It would be helpful to understand the choice of sample geometry used (~12 mm radius sphere) - is this the ideal one for ZULF NMR or is it constrained by other considerations such as the SABRE hyperpolarization or the size of the OPM? In Section V, the authors claim that the "smaller sensor size enables investigation of smaller samples" but there is a tension here. Smaller sample volume means fewer spins and therefore lower SNR; the diamond sensor already struggles with SNR as compared to the OPM. Is this claim simply because the OPM cannot get closer than 11.8 mm, whereas the NV could get much closer to a very small sample? Could the bandwidth of the OPM be extended to higher frequencies?

It would also be helpful to see the comparison between the NV and the OPM data in terms of signal per unit time. How important is the SABRE hyperpolarization to achieving NV based measurements? What are the prospects for measuring samples that cannot be hyperpolarized via para-hydrogen techniques - and need to use standard high-field Boltzmann prepolarization techniques? Without addressing this question, the claim in the last sentence of the abstract is not really substantiated.

In addition to the broad considerations above, the authors also need to improve the quality of the writing in the manuscript. There are several errors, and the arguments are occasionally hard to follow. The authors also need to expand on the details so that other researchers in the field have a clearer understanding of the technique. For example, the authors indicate they are using the methods shown in references 19 and 23, but the actual experimental conditions are missing from the current manuscript. How is the extraction of the J-couplings done? Does it require a priori knowledge of the molecular structure?

Specific Comments

1. Figure 2 is hard to read for a number of reasons. The red and pink dots, for example, are really hard to see, and in fact hard to find. The authors should also label the sample itself and consider making the figure larger and more readable.
2. In Figure 3, the authors should comment on the NV data near zero frequency ($1/f$ noise?). They should also comment on the increased disagreement between the different experiments and the data at 40 nT - particularly in the region around 1 Hz.
3. Figures 4 a,b,c need x-axis labels. The authors should also discuss the linewidth changes with field.
4. There is almost no discussion of Figure 5 in the main text. The interpretation of Figure 5a is not clear. The OPM signal is argued to fall along the expected scaling for a dipole source. But this data is not measured (probably because it cannot be measured closer) below 11.8 mm; the data could also fall along a straight line. The diamond spectra area appears to fall along a straight line, as well (measured to much closer distance), which the authors recognize is because the spectra are subjected to a magnetic field gradient over the volume of the sample. Could it be the case that the OPM is also sampling this gradient, but at larger distances? In that case, we expect the signal to scale as $1/r^4$, which we do not observe for either sample.

Reviewer #4

(Remarks to the Author)

I co-reviewed this manuscript with one of the reviewers who provided the listed reports. This is part of the Communications Chemistry initiative to facilitate training in peer review and to provide appropriate recognition for Early Career Researchers who co-review manuscripts.

Version 1:

Reviewer comments:

Reviewer #2

(Remarks to the Author)

The authors have addressed the comments raised by me except the one given below:

Page 4, 3rd Paragraph: The authors mention "Although this demodulated signal is magnetically insensitive to first order (as discussed earlier)". It is not clear to me in which section was it discussed earlier. In fact, the entire paragraph is not clear to me. The authors should support their statements by analytical expressions and they should be included in the Methods section.

The analytical expressions in the Methods section is still not included.

Reviewer #3

(Remarks to the Author)

The resubmitted manuscript is much improved and has largely addressed our concerns. We recommend publication after the authors address the following concerns that remain.

1. There are still some issues with the flow and organization of the manuscript. For example, there is almost no reference to Figure 1 in the main text, with most of the discussion starting with Figure 2.
2. The paper could benefit from having additional experimental details added to the Appendix. As written, it would be very hard, if not impossible, for another group to duplicate their experiments. This is very important to maintain credibility of the scientific enterprise.

Reviewer #4

(Remarks to the Author)

I co-reviewed this manuscript with one of the reviewers who provided the listed reports. This is part of the Communications Chemistry initiative to facilitate training in peer review and to provide appropriate recognition for Early Career Researchers who co-review manuscripts.

We thank the editors and referees for their constructive comments. We believe the revised manuscript, which is attached along with a list of changes, addresses the referee comments. Below we include point-by-point detailed responses (in black) to the referees questions and comments (in blue) and relevant changes to the text (in green).

Reviewer #1 (Remarks to the Author):

Reviewer comment:

The manuscript entitled 'Zero- to low-field J-spectroscopy with a diamond magnetometer' demonstrates the advantages of diamond magnetometers for conducting ZULF (Zero- to ultralow-field) NMR spectroscopy. Most of results are about the comparisons between a commercial optically pumped magnetometer (OPM) and a home-built diamond magnetometer. Obviously, OPM provides a way higher SNR due to its high-sensitivity. However, the diamond magnetometer has several advantages over OPM in terms of bandwidth and detection volume. The results sound reasonable. The design of the home-built diamond magnetometer looks highly practical.

Response :

We thank the reviewer for carefully reading and accurately summarizing the manuscript.

Reviewer comment:

This work is indeed the first demonstration of NV diamond detected ZUFL-NMR. However, I doubt that it actually demonstrates significant advances. The feature that NV diamond can do sensing at over 500 Hz seems a minor factor, compared to the OPMs high-sensitivity. To achieve the same SNR as that with OPM, NV diamond may need hundreds of scans, which is certainly a significant disadvantage.

Response:

Fair enough. However, the inferior sensitivity is partially compensated by the possibility of smaller standoff distances of the sensor which is a general motivation for using NV magnetometer as opposed to OPMs. It should also be pointed out that while sensitivity is generally important, efficient hyperpolarization techniques can in certain applications relax the sensitivity requirements.

We sharpened the discussion of this point in several positions of the manuscript. For example:

Changes in the manuscript :

The demonstration of ZULF NMR in this manuscript confirms this possibility, once the sensor's residual magnetic field effects are under control. These can be expected to be less of a problem for smaller samples. Sample volumes on the order of the diamond sensor dimensions are possible to detect, since the surface magnetic field of a uniformly magnetized spherical sample is independent of its radius r . The field outside the sample is equivalent to the field of a magnetic point dipole at the center of the sample with a magnetic moment proportional to the number of spins. For a constant density, the number of spins scales with the volume (r^3) while the signal of a dipole measured at position r reduces like r^{-3} . Therefore, in the regime when the sample magnetization can be treated a classical dipole, miniaturization of the sensor permits probing smaller samples without a fundamental loss in sensitivity.

Reviewer comment:

Moreover, the demonstration of the 2nd advantage that NV diamond can detect small volume samples is not clear in this manuscript. The authors could have used unconventional

samples with sub-millimeter scales such as micro-fluids, where OPM detected ZULF NMR is nearly impractical.

Response:

Indeed, we plan to go in this direction of micro-fluidic systems in future work. The focus of this paper is the first demonstration of ZULF NMR detection with an NV center-based magnetometer. We decided to start with an NMR sample that has a large signal detectable with the OPM as well. We feel that the current results are already valuable for the community, indicating convincingly that the path towards smaller samples is open.

Reviewer comment:

In addition, I noticed that some of the important technical concepts are not clearly described in the manuscript. Thus, considering the scientific impact of the manuscript, I don't recommend it for publication in Communications in Chemistry.

Response:

We had attempted to present sufficient technical details in the original manuscript, however based on this and other Referees' remarks we have revised and expanded the technical descriptions substantially, and we hope that the Referee finds the current revised version satisfactory.

Reviewer comment:

1. The authors stated they used a modulated bias protocol for the operation of their diamond sensor for zero-field magnetometry. But, it is rather hard to understand the working-principle and real implementation of this protocol. The authors should include a more detail description on it.

Response:

We thank the reviewer for this comment and amended the manuscript accordingly.

Changes in the manuscript :

In this study, we employ a zero-field magnetometry protocol with a modulated bias (6.12 kHz along [100] crystallographic direction with μT amplitude) achieving sensitivities of $13 \text{ pT}/\sqrt{\text{Hz}}$ comparable to magnetically biased ODMR^{23,28}. The presence of a μT -scale magnetic field broadens the zero-field ODMR feature, which, in turn, reduces contrast. This change in peak amplitude can itself be viewed as a form of magnetic resonance, with a peak at zero bias field utilizing all NV axes. Therefore, by modulating and demodulating the magnetic field, a magnetic dispersive signal can be extracted. This signal features a zero crossing at zero bias, allowing for high-sensitivity sensing with a linear response in this regime. This method was already demonstrated and described analytically as an alternative to zero-field sensing using circularly polarized microwaves²³.

Reviewer comment:

2. There are several wrong citation of figures. For example, in page 4, "the diamond sensor compared to the OPM sensor is presented in Fig. 3", Fig.3 -> Fig. 4. In page 5, "the advantage of the higher bandwidth of the diamond sensor is in Fig. 3.", Fig.3 -> Fig.4.

Response:

We sincerely apologize for the mix-up with figure references and that has occurred during multiple rounds of editing of the manuscripts. The references have now been fixed and double checked.

Reviewer comment:

3. Please clarify the materials that the authors used for NV diamond sensor and the potential source of the residual magnetic field gradient.

Response:

We added in the appendix details related to our study of what magnetic components could be the source of the gradient observed in the spectra when the diamond sensor was moved closer to the NMR tube.

Reviewer comment:

4. Please explain why with NV diamond sensor the ^{15}N NMR spectra were invisible.

Response:

As can be seen in Figure 4, the signal sizes of ^{15}N NMR signals are approximately 1 pT in amplitude. Unfortunately, this is within the noise floor of the averaged diamond sensor signal. We added a more detailed explanation of it. We also included the diamond sensor data in the spectra and analysed a histogram of the diamond sensor noise in Figure 4.

Changes in the manuscript:

In the diamond data no ^{15}N precession features are resolvable. We relate this due to the sensitivity difference of the sensors. This can be further illustrated in two, ideally complementary, ways. Both the sensitivity characterized by the amplitude spectral density (ASD) as well as the histogram of the amplitude distribution of the diamond sensor data will be in the following used to discuss this limitation quantitatively. The ASD of the diamond sensor noise is displayed in Fig. 4 i). We measure 9 s time traces of either magnetically sensitive (i.e. with magnetic field modulation and pump light applied), insensitive (i.e. with only pump light applied) and dark (i.e. with neither pump light nor magnetic field modulation applied) diamond data and apply a Fourier transform to extract the ASD as a characterization of the magnetic sensitivity. The magnetic sensitivity of the diamond sensor given by the noise baseline of the magnetically sensitive ASD of the sensor is measured to be 13 pT/ $\sqrt{\text{Hz}}$ (average within the range: 20 Hz to 30 Hz). The expected amplitude spectrum root means square (RMS) value for the single-sided ASD of 13 pT/ $\sqrt{\text{Hz}}$, considering the applied Hann window before the Fourier transform, 9 s measurement time and 50 averages, is 0.8 pT. In Fig. 4 h), we show the histogram of the diamond sensor amplitude spectrum between 26 Hz and 35 Hz. The histogram data is Rayleigh distributed and fitted accordingly. A Rayleigh distributed histogram is expected for the absolute value of the single sided amplitude spectrum of normally distributed random time series data. The fitting parameter of the Rayleigh distribution σ is 1.4 pT. The corresponding RMS is $\sqrt{2}\sigma \approx 2$ pT, larger than the ^{15}N -related precession feature amplitude.

Reviewer #2**Reviewer comment:**

Muhib Omar et al. report on low-field detection of nuclear magnetic resonance with ensembles of nitrogen vacancy centers. They focus their attention on the measurement of the ZULF-NMR spectrum at (or close to) zero magnetic field, and together with the comparison with the OPM measurements, they are able to validate the multi-peak spectrum resulting from a combination of J-coupling and small bias magnetic fields. Not only that the manuscript has quite a few novelties, its high quality and scientific rigor merit publication in COMMSCHEM. I really enjoyed reading it.

Response:

We thank the Referee for carefully reading the manuscript, the accurate summary and for positively evaluating our work.

Reviewer comment:

However, I would like to point out to the authors that there are many sloppy errors in the manuscript which indicates that the manuscript was not carefully proofread by the authors before submission. This will increase the time the reviewers have to spend to understand all the details.

Response:

We sincerely apologize for the sloppy errors that penetrated the manuscript during the multiple rounds of editing it. We have now attempted to clean up and carefully check the manuscript.

Reviewer comment:

1. Since the diamond sensor, flexible PCB and other components of the experimental setup used in this work were not described in previous papers, a detailed description containing the part numbers of the components used should be included in the section IIIB. An actual photograph of the diamond sensor setup should be included in the Supporting Information document.

Response:

We thank the reviewer for this constructive comment. We added a full description of the components of the diamond sensor and the PCBs in appendix A and included pictures of the sensor close to the sample, the sensor, and the sensor head. We found this more suitable than in section 3 B, considering the amount of material we added and the readability of the manuscript.

Reviewer comment:

2. A Table summarizing the comparison of diamond magnetometer with other ZULF-NMR sensors such as OPMs, SQUIDs and magnetoresistive sensors etc. should be included. The parameters to be considered are sensitivity, dynamic range, bandwidth, stand-off distance, blocked optical access etc. The corresponding References should also be cited in the Table.

Response:

We added this table in the introduction on page 2 with the exception of the blocked optical access as we found it difficult to quantify.

Reviewer comment:

3. Page 4, 3rd Paragraph: The authors mention "Although this demodulated signal is magnetically insensitive to first order (as discussed earlier)". It is not clear to me in which section was it discussed earlier. In fact, the entire paragraph is not clear to me. The authors should support their statements by analytical expressions and they should be included in the Methods section.

Response:

We added a more detailed description of the protocol itself at the end of section 2.

Changes in the manuscript :

While the zero-field feature frequency is independent to magnetic field changes around zero bias-magnetic fields, temperature variations still result in frequency shifts²⁹. To avoid systematic effects due to these shifts, we stabilize the MW to the zero-field feature. For this we generate an error signal by sinusoidally frequency modulating the MW frequency (modulation frequency 140 kHz, modulation amplitude 100 kHz) and demodulating it with a lock-in amplifier. The demodulated signal is linear to frequency shifts due to temperature variation and just quadratic to frequency shifts due to magnetic field changes. Stabilizing the MW frequency via a proportional-integral-differential (PID) control loop makes the system largely insensitive to temperature fluctuations, enabling long-term measurement campaigns.

Reviewer comment:

4. Please clarify why the Fourier transform (FT) is applied after 50 averages in the time domain. Why not average the 50 FTs?

Response:

Mathematically, averaging in the time domain or averaging the corresponding Fourier transforms is equivalent because the Fourier transform is a linear operation. We therefore do not expect any difference in signal-to-noise ratio between the two approaches.

Reviewer comment:

5. Please clarify in the text why “the ^{15}N resonances in acetonitrile were detectable only with the OPM.” using sensitivity calculation arguments, if possible.

Response:

This is an important point also raised by other reviewers. We added the diamond data in the plots illustrating that the noise level was simply too high to detect the ^{15}N resonances. We added a more detailed explanation of it. We also included the diamond sensor data in the spectra and analysed a histogram of the diamond sensor noise in Figure 4.

Changes in the manuscript:

In the diamond data no ^{15}N precession features are resolvable. We relate this due to the sensitivity difference of the sensors. This can be further illustrated in two, ideally complementary, ways. Both the sensitivity characterized by the amplitude spectral density (ASD) as well as the histogram of the amplitude distribution of the diamond sensor data will be in the following used to discuss this limitation quantitatively. The ASD of the diamond sensor noise is displayed in Fig. 4 i). We measure 9 s time traces of either magnetically sensitive (i.e. with magnetic field modulation and pump light applied), insensitive (i.e. with only pump light applied) and dark (i.e. with neither pump light nor magnetic field modulation applied) diamond data and apply a Fourier transform to extract the ASD as a characterization of the magnetic sensitivity. The magnetic sensitivity of the diamond sensor given by the noise baseline of the magnetically sensitive ASD of the sensor is measured to be $13 \text{ pT}/\sqrt{\text{Hz}}$ (average within the range: 20 Hz to 30 Hz). The expected amplitude spectrum root means square (RMS) value for the single-sided ASD of $13 \text{ pT}/\sqrt{\text{Hz}}$, considering the applied Hann window before the Fourier transform, 9 s measurement time and 50 averages, is 0.8 pT. In Fig. 4 h), we show the histogram of the diamond sensor amplitude spectrum between 26 Hz and 35 Hz. The histogram data is Rayleigh distributed and fitted accordingly. A Rayleigh distributed histogram is expected for the absolute value of the single sided amplitude spectrum of normally distributed random time series data. The fitting parameter of the Rayleigh distribution σ is 1.4 pT. The corresponding RMS is $\sqrt{2}\sigma \approx 2 \text{ pT}$, larger than the ^{15}N -related precession feature amplitude.

Reviewer comment:

6. If I am not wrong, the data in most of the subfigures of the Figure 4 is not consistent with the description given in the caption. Please double-check.

Response:

We thank the reviewer for drawing our attention to these mistakes and corrected the caption of Figure 4.

Reviewer comment:

7. Please specify the sources of the “1/f”-like magnetic noise in the Figure 4(i) and how the noise can be mitigated. Please cite the references, if possible.

Response:

The main source of 1/f noise is likely the current source used for the field modulation in the sensing protocol. We added the following discussion on page 7 at the last full paragraph in the left column.

Changes in the manuscript:

Currently, the magnetic noise floor of the sensor [in Fig. 4 i)] is dominated by 1/f noise at frequencies below 5 Hz (for 9 s data traces). A relevant frequency range, since some of the observed NMR signals occur there. The magnetically insensitive noise trace does not show this low-frequency noise, indicating that neither laser nor electronic noise is responsible. Environmental magnetic noise can also be excluded, as the OPM sensor does not exhibit comparable features (see Fig. 9). Temperature effects that could mimic such a noise increase are mitigated using a microwave frequency lock. Therefore, a plausible explanation is that the observed 1/f noise originates magnetically from the field modulation source of the diamond sensing protocol. This is inactive during OPM operation. The DC component of the source is necessary to calibrate the sensitivity of the sensor by applying constant fields of known amplitude. In a future iteration of this sensor, the calibration protocol will be modified and the modulation coil current high-pass filtered, effectively removing this 1/f-noise contribution.

Reviewer comment:

8. Please specify in the text the details of the equivalent noise bandwidth and the hardware low-pass filtering used for the diamond sensor.

Response:

The parameters were added in the diamond sensor section on page 4.

Reviewer comment:

9. "The sensitivity was sufficient to resolve the characteristic lines at zero field in the diamond data related to 1H-15N nuclei in isotopically labeled acetonitrile within 50 averages." I got an impression that 15N resonances are not detectable?

Response:

The signal due to the j-coupled 1H-15N nuclei were detectable with the diamond sensor within 50 Hz. The signal amplitude is around 35 pT in amplitude and displayed in most plots of the diamond sensor data. What the diamond sensor could not resolve were the 15N precession resonances for applied background fields of 7 μ T and 14 μ T. These signals are just 1 pT in amplitude and only observable with the OPM. There were several comments related to the 15N resonances and we sharpened and improved the manuscript in this regard.

We added the diamond data in the plots illustrating that the noise level was simply too high to detect the 15N resonances. We added a more detailed explanation of it. We also included a histogram of the diamond sensor noise in Figure 4. Overall we tried to be more rigorous about the terminology regarding the different resonances. We added the following text:

Changes in the manuscript:

In the diamond data no 15N precession features are resolvable. We relate this due to the sensitivity difference of the sensors. This can be further illustrated in two, ideally complementary, ways. Both the sensitivity characterized by the amplitude spectral density (ASD) as well as the histogram of the amplitude distribution of the diamond sensor data will be in the following used to discuss this limitation quantitatively. The ASD of the diamond sensor noise is displayed in Fig. 4 i). We measure 9 s time traces of either magnetically sensitive (i.e. with magnetic field modulation and pump light applied), insensitive (i.e. with only pump light applied) and dark (i.e. with neither pump light nor magnetic field modulation

applied) diamond data and apply a Fourier transform to extract the ASD as a characterization of the magnetic sensitivity. The magnetic sensitivity of the diamond sensor given by the noise baseline of the magnetically sensitive ASD of the sensor is measured to be $13 \text{ pT}/\sqrt{\text{Hz}}$ (ASD average within the range: 20 Hz to 30 Hz). The expected amplitude spectrum root means square (RMS) value for the single-sided ASD of $13 \text{ pT}/\sqrt{\text{Hz}}$, considering the applied Hann window before the Fourier transform, 9 s measurement time and 50 averages, is 0.8 pT. In Fig. 4 h), we show the histogram of the diamond sensor amplitude spectrum between 26 Hz and 35 Hz. The histogram data is Rayleigh distributed and fitted accordingly. A Rayleigh distributed histogram is expected for the absolute value of the single sided amplitude spectrum of normally distributed random time series data. The fitting parameter of the Rayleigh distribution σ is 1.4 pT. The corresponding RMS is $\sqrt{2}\sigma \approx 2 \text{ pT}$, larger than the ^{15}N -related precession feature amplitude.

Reviewer comment:

Minor comments and corrections:

1. Section IIIA, 1st Paragraph, 2nd Sentence: “prior to” should be more appropriate. The same sentence, which ends with both full stop and a comma, should be corrected.

Response:

We thank the reviewer for the suggestion and implemented it.

Reviewer comment:

2. Green laser excitation power used for ZULF NMR measurements should be mentioned.

Response:

We used 150mW of laser light on the diamond. We added this information in the text.

Reviewer comment:

3. Page 4, Paragraph 4: “We average the noise floor between 5 and 30 Hz.” This needs to be corrected.

Response:

We thank the reviewer for the correction and implemented it.

Reviewer comment:

4. Page 4, Paragraph 4: “turning off” instead of “turning of”.

Response:

We thank the reviewer for the correction and implemented it.

Reviewer comment:

5. The frequency of the modulated bias field and the MW field should be mentioned.

Response:

We thank the reviewer for the correction and mention both values in the text.

Reviewer comment:

6. The title for the inset of the Figure 3 is not completely visible.

Response:

We corrected the figure.

Reviewer comment:

7. Figure 3: Dashed lines instead of dotted lines.

Response:

We thank the reviewer for the correction and implemented it.

Reviewer comment:

8. The description of the Figures in the main text and the corresponding Figure numbers are not consistent with each other at several places in the manuscript.

Response:

We thank the reviewer for spotting this mistake, corrected it and double checked the referencing.

Reviewer comment:

9. Page 4, Section IIID: "For observing the nuclear spin precession along the y-axis". Please specify the plane in which the transverse nuclear magnetization precesses.

Response:

We added this information in the single paragraph of section 3D .

Reviewer comment:

10. Please double-check for any errors in the caption of figure 5.

Response:

We double checked, corrected the mistakes and reformulated part of the caption for clarity.

Reviewer comment:

11. Section V, 1st Paragraph: Please check for the presence of any errors.

Response:

We checked and corrected our errors there.

Reviewer comment:

I request the authors to address the above points in the revised version of the manuscript. The manuscript can be sent to me for review again.

Response:

We addressed all the points of the reviewer and believe to have substantially improved the manuscript. We appreciate the willingness to review the manuscript again.

Reviewer #3 (Remarks to the Author):

Reviewer comment:

The paper is a demonstration of zero/ultra-low field (ZULF) NMR magnetometry with nitrogen-vacancy (NV) centers in diamond, as a means of measuring J-coupling strengths. The authors built an apparatus to perform ZULF NMR spectroscopy with the NV center as a probe and compared the NV magnetometry results to results obtained through commercial techniques. The authors examine [15N]-acetonitrile using PHIP-SABRE (a para-hydrogen based nuclear spin hyperpolarization technique) and measure the hyperpolarized J-coupling signal from the sample using ODMR. The authors compare the efficacy of the diamond sensor to a standard vapor-cell-based optically pumped magnetometer (OPM) method, and demonstrate that though the diamond sensor does not have a comparable noise floor to the OPM, it can currently be useful at frequencies above the low-pass cutoff of the OPM.

Response:

We thank the Referee for the accurate summary

Reviewer comment:

This is a very nice demonstration of NV-based NMR in the ZULF regime. This novel result, and the comparison with OPM methods could be of interest to the community and the wider field. It is a little harder to gauge the significance of the results presented. Generally, NV magnetometry is known to have better spatial resolution but worse magnetic field sensitivity compared to an OPM. The comparison between the NV and OPM platform ideally needs to be performed in the context of the target application. It would be helpful to understand the choice of sample geometry used (~12 mm radius sphere) - is this the ideal one for ZULF NMR or is it constrained by other considerations such as the SABRE hyperpolarization or the size of the OPM?

Response:

We appreciate the positive assessment of our work. While a spherical shape is not the shape that maximizes the signal it has many practical advantages, such as the absence of geometry dependent magnetic fields on the sample due to the sample magnetization, it minimizes surface area per volume, which is important to slow down sample evaporation, is straightforward to be manufactured and can withstand the pressure during the bubbling of parahydrogen. The possible reduction of sample volume, which, as the reviewer points out, is one of the advantages of NV sensors, is planned for future work.

Reviewer comment:

In Section V, the authors claim that the "smaller sensor size enables investigation of smaller samples" but there is a tension here. Smaller sample volume means fewer spins and therefore lower SNR; the diamond sensor already struggles with SNR as compared to the OPM.

Is this claim simply because the OPM cannot get closer than 11.8 mm, whereas the NV could get much closer to a very small sample?

Response:

This is just true if the noise is limited by the spin noise of the sample. The surface field of a uniformly magnetized sphere is independent of the radius (the number of spins goes as r^3 and the dipole field drops as $1/r^3$), therefore, if the detection is not limited by spin noise, a smaller sensor allows to go to smaller samples without loss of signal-to-noise. This is an important point of the manuscript and we added the following in the discussion section.

Changes in the manuscript:

The demonstration of ZULF NMR in this manuscript confirms this possibility, once the sensor's residual magnetic field effects are under control. These can be expected to be less of a problem for smaller samples. Sample volumes on the order of the diamond sensor dimensions are possible to detect, since the surface magnetic field of a uniformly magnetized spherical sample is independent of its radius r . The field outside the sample is equivalent to the field of a magnetic point dipole at the center of the sample with a magnetic moment proportional to the number of spins. For a constant density, the number of spins scales with the volume (r^3) while the signal of a dipole measured at position r reduces like r^{-3} . Therefore, in the regime when the sample magnetization can be treated a classical dipole, miniaturization of the sensor permits probing smaller samples without a fundamental loss in sensitivity.

Reviewer comment:

Could the bandwidth of the OPM be extended to higher frequencies?

Response:

In principle, yes. However, extending the bandwidth from DC comes with a loss of sensitivity and additional technical challenges. Ultimately both systems detect the response of electron spins to magnetic fields. The differences are spin density and coherence time. For OPMs, the intrinsically better coherence time can be traded for bandwidth, however, the spin density for NV centers is higher than for OPMs, so if both have the same coherence time, the diamond sensor is more sensitive. A detailed discussion of these topics goes beyond the scope of a paper devoted to NV magnetometry. Commercial OPMs that meet the necessary sensitivity and bandwidth specifications are not currently available.

Reviewer comment:

It would also be helpful to see the comparison between the NV and the OPM data in terms of signal per unit time.

Response:

This was a common remark by the reviewer. We added a table illustrating the different sensing modalities considering their sensing parameters. We included examples of other modalities as well that were used for ZULF NMR to give an overview.

Reviewer comment:

How important is the SABRE hyperpolarization to achieving NV based measurements?

Response:

While traditional high-field NMR relies on thermal polarization of nuclear spins, ZULF NMR necessarily requires polarization of some kind. SABRE techniques are currently among the ones producing highest molar polarization and this is why we chose it to demonstrate ZULF NMR for the first time with a diamond magnetometer.

Reviewer comment:

What are the prospects for measuring samples that cannot be hyperpolarized via para-hydrogen techniques - and need to use standard high-field Boltzmann prepolarization techniques? Without addressing this question, the claim in the last sentence of the abstract is not really substantiated.

Response:

Indeed, a number of hyperpolarization techniques are used in our laboratory including: prepolarization in high-field by sample shuttling, dynamical nuclear polarization (DNP), photo-induced DNP and a variety of parahydrogen-based techniques

As mentioned above, SABRE currently provides the highest molar polarization, motivating its use in this work. The different polarization techniques are now discussed in the discussion section with the following addition:

Changes in the manuscript:

In the following we discuss the generalization of our results to other ZULF NMR samples. Firstly, at zero field a source of polarization is always required to produce detectable signals². Among the various techniques available⁴², Signal Amplification by Reversible Exchange (SABRE) is selected in this study due to its ability to produce high molar polarization under ambient conditions. However, the methodology presented here is not limited to SABRE. One can also employ several complementary hyperpolarization methods, including high-field prepolarization followed by sample shuttling, dynamic nuclear polarization (DNP), photo-induced DNP, and other parahydrogen-based approaches. While

SABRE enables the first demonstration to our knowledge of diamond-based ZULF NMR, these alternative strategies broaden the scope of detectable molecular targets, including those not amenable to parahydrogen-based hyperpolarization. Thus, we view the use of NV sensors in conjunction with diverse hyperpolarization techniques as a viable platform for scalable and versatile low-field NMR detection for a variety of polarizable compounds.

Reviewer comment:

In addition to the broad considerations above, the authors also need to improve the quality of the writing in the manuscript. There are several errors, and the arguments are occasionally hard to follow.

Response:

We apologize for the poor writing and proofreading. We have performed major revisions and believe to have substantially improved the manuscript.

Reviewer comment:

The authors also need to expand on the details so that other researchers in the field have a clearer understanding of the technique. For example, the authors indicate they are using the methods shown in references 19 and 23, but the actual experimental conditions are missing from the current manuscript.

Response:

We have added substantial amounts of detail to the manuscript.

For instance:

Changes in the manuscript:

In this study, we employ a zero-field magnetometry protocol with a modulated bias (6.12 kHz along [100] crystallographic direction with μT amplitude) achieving sensitivities of $13 \text{ pT}/\sqrt{\text{Hz}}$ comparable to magnetically biased ODMR^{22,27}. The presence of a μT -scale magnetic field broadens the zero-field ODMR feature, which, in turn, reduces contrast. This change in peak amplitude can itself be viewed as a form of magnetic resonance, with a peak at zero bias field utilizing all NV axes. Therefore, by modulating and demodulating the magnetic field, a magnetic dispersive signal can be extracted. This signal features a zero crossing at zero bias, allowing for high-sensitivity sensing with a linear response in this regime. This method was already demonstrated and described analytically as an alternative to zero-field sensing using circularly polarized microwaves²². While the zero-field feature frequency is independent to magnetic field changes around zero bias-magnetic fields, temperature variations still result in frequency shifts²⁸. To avoid systematic effects due to these shifts, we stabilize the MW to the zero-field feature. For this we generate an error signal by sinusoidally frequency modulating the MW frequency (modulation frequency 140 kHz, modulation amplitude 100 kHz) and demodulating it with a lock-in amplifier. The demodulated signal is linear to frequency shifts due to temperature variation and just quadratic to frequency shifts due to magnetic field changes. Stabilizing the MW frequency via a proportional-integral-differential (PID) control loop makes the system largely insensitive to temperature fluctuations, enabling long-term measurement campaigns.

Reviewer comment:

How is the extraction of the J-couplings done? Does it require a priori knowledge of the molecular structure?

Response:

The extraction is done by a Fourier transform of the time series data acquired by the diamond magnetometer and fitting of the spectra with Lorentzians. The corresponding peaks of the XA3 system are expected to be at J and 2J at zero bias field. In the current

demonstration experiment, we have a priori knowledge of the molecule under study. However, the spectra of unknown substances can also be measured. The observed peak number, peak positions and behaviour under applied magnetic fields can be used to identify the substance.

Reviewer comment:

Specific Comments

1. Figure 2 is hard to read for a number of reasons. The red and pink dots, for example, are really hard to see, and in fact hard to find. The authors should also label the sample itself and consider making the figure larger and more readable.

Response:

We thank the reviewer for the editorial suggestion and increased the dot size, added the sensor label in it and enlarged the figure altogether. We believe it has improved.

Reviewer comment:

2. In Figure 3, the authors should comment on the NV data near zero frequency (1/f noise?). They should also comment on the increased disagreement between the different experiments and the data at 40 nT - particularly in the region around 1 Hz.

Response:

We ascribe the 1/f noise-like excess to the actual 1/f noise of the diamond sensor. This can be seen as well in Figure 4i) but is more pronounced in this data set since it is of longer duration. The disagreements between OPM and diamond data with respect to the simulation can be due to experimental imperfections, for example the observed field inhomogeneity of the piercing solenoid. And we also observe in other datasets, that the diamond sensor affects the spectra (broadening them). Furthermore, the magnetic noise observable in the diamond sensor trace might well be a result from the diamond sensor field modulation which in turn then affects the NMR spectra and makes it in general more complicated. We added this statement in the caption of Fig. 3.

Changes in the manuscript:

The minor disagreement between simulation and experimental data can be explained with experimental imperfections like magnetic field gradients of the applied fields and the noise properties of the sensors, in particular the 1/f noise of the diamond sensor.

Reviewer comment:

3. Figures 4 a,b,c need x-axis labels. The authors should also discuss the linewidth changes with field.

Response:

We thank the reviewer for the editorial suggestion and improved the figure. The broadening increases most likely due to the inhomogeneity of the applied solenoid field. We added this in the text:

Changes in the manuscript:

We observe broadening of the spectra as a function of the applied field. We relate this to inhomogeneities of the piercing solenoid field.

Reviewer comment:

4. There is almost no discussion of Figure 5 in the main text. The interpretation of Figure 5a is not clear. The OPM signal is argued to fall along the expected scaling for a dipole source. But this data is not measured (probably because it cannot be measured closer) below 11.8 mm; the data could also fall along a straight line. The diamond spectra area appears to fall

along a straight line, as well (measured to much closer distance), which the authors recognize is because the spectra are subjected to a magnetic field gradient over the volume of the sample. Could it be the case that the OPM is also sampling this gradient, but at larger distances? In that case, we expect the signal to scale as $1/r^4$, which we do not observe for either sample.

Response:

We have actually reanalysed the data of Fig. 5a), included a description of the analysis method and expanded the discussion quite a lot. We believe that we have a good understanding of what is going on and the data seem to suggest that both data sets are consistent with $1/r^3$ scaling up to some distance. We added a substantial amount of discussion regarding the residual disagreement. The following statements were added:

Changes in the manuscript:

The diamond-detected spectra for close sample-sensor distances appear broadened. Surprisingly, with both sensors operating under identical conditions around the NMR sample, the OPM data appear far less broadened. Both OPM and diamond sensor are turned off when the other sensor is in operation to avoid interference from each sensor's modulating magnetic field. Broadening of NMR spectra can be related to varying transverse magnetic fields during the measurement time and/or magnetic field gradients over the sample. Possible sources of magnetic fields and gradients are investigated. We find that the diamond sensor components are slightly magnetic. Spectra measured with an OPM for different diamond sensor components placed close to the NMR sample can be found in Appendix B. The broadening could also be caused by slow magnetic field drifts of the modulation source that could affect the NMR spectra. The interpretation of the broadening is complicated by FIG. 5. Distance scaling of the NMR spectra. Distance is given from sensor volume center to sample volume center. The sample is a glass sphere with a 6.25 mm radius surrounded by a piercing solenoid with a 0.5 mm wall thickness. a) Normalized signal strength of the 2J-peak measured at different distances with respect to the NMR sample for the diamond (gold) and OPM (blue) sensor. The signal strength is the baseline subtracted numerical integral of the spectra between 2.5 Hz and 6 Hz capturing the 2J peak for the respective sensor. For each sensor, the signal strength is normalized to the value at the distance of 14.3 mm. The closest distance possible for the OPM sensor is 11.8 mm. The closest distance for the diamond sensor is 6.8 mm, dominated by the diameter of the piercing solenoid. The normalized signal strength is compared to the expected scaling for a dipole source (black). b) Sketch of the experiment. c) Spectra recorded using the OPM for various distances. d) Spectra recorded using the NV sensor for various distances. NV data are 50 times averaged in the time domain before Fourier transforming. The spectra in c) and d) are offset as a visual aid. Note, that the OPM data are recorded with natural abundance acetonitrile leading to the much smaller signal size compared to the diamond data. The shimming procedure used during the sensor distance experiment. Shimming is conducted for each diamond sensor position using the OPM sensor recorded spectra as a reference. This is necessary to compensate the effect of the slight magnetic field components of the diamond sensor. In a future iteration of the experiment, reducing the magnetic influence of the diamond sensor using less magnetic components and a high-pass filtered current source might resolve that discrepancy.

Reviewer #4 (Remarks to the Author):

I co-reviewed this manuscript with one of the reviewers who provided the listed reports. This is part of the Communications Chemistry initiative to facilitate training in peer review and to provide appropriate recognition for Early Career Researchers who co-review manuscripts.

Response:

We thank the reviewer for co-reviewing the manuscript and support the initiative. The review process is essential for scientific culture and training improves the peer review process.

We thank the editors and referees for their constructive comments. We believe the revised manuscript, which is attached along with a list of changes, addresses the referee comments. Below we include point-by-point detailed responses (in black) to the referees questions and comments (in blue) and relevant changes to the text (in green).

Reviewer #2 (Remarks to the Author):

Reviewer comment:

The authors have addressed the comments raised by me except the one given below:

Page 4, 3rd Paragraph: The authors mention “Although this demodulated signal is magnetically insensitive to first order (as discussed earlier)”. It is not clear to me in which section was it discussed earlier. In fact, the entire paragraph is not clear to me. The authors should support their statements by analytical expressions and they should be included in the Methods section.

The analytical expressions in the Methods section is still not included.

Response :

We thank the reviewer for revisiting the manuscript. We apologize for the earlier misunderstanding of the request and have now added a subsection in the Methods section that offers a more detailed analytical explanation of the MW demodulation and its purpose.

Changes in the manuscript:

On page 3 , Methods B : Starting from “In the following we discuss the microwave locking technique and the zero-field sensing methodology. ...” until second paragraph of page 4.

Reviewer #3 (Remarks to the Author):

Reviewer comment:

The resubmitted manuscript is much improved and has largely addressed our concerns. We recommend publication after the authors address the following concerns that remain.

1. There are still some issues with the flow and organization of the manuscript. For example, there is almost no reference to Figure 1 in the main text, with most of the discussion starting with Figure 2.

Response :

We thank the reviewer for examining the manuscript again. We added a discussion of figure 1 in the beginning of the Methods section.

Changes in the manuscript :

On page 3, Methods A: In the absence of a bias field, the main frequency components of the signal are determined by the J-coupling constant of -1.69 Hz (Fig. 1a), resulting in two principal transitions within the zero-bias energy-level structure (see Fig. 1b) of the compound: one at the J-frequency and the other at twice that

frequency. These transitions can be magnetically measured and are visible in the amplitude spectrum of the magnetization decay of the sample (Fig. 1c).

Reviewer comment:

2. The paper could benefit from having additional experimental details added to the Appendix. As written, it would be very hard, if not impossible, for another group to duplicate their experiments. This is very important to maintain credibility of the scientific enterprise.

Response :

We appreciate the reviewer's suggestion to expand the experimental detailing to enable reproducibility. We added additional details throughout the manuscript to make clear what devices were used.

We hope that with these corrections, the manuscript can be accepted for publication in Communications Chemistry.

Sincerely,
Muhib Omar on behalf of the authors